# Learning Algorithms for Markovian Bandits:
# Is Posterior Sampling more Scalable than Optimism?

**Nicolas Gast**                                                   *nicolas.gast@inria.fr*
**Bruno Gaujal**                                                   *bruno.gaujal@inria.fr*
**Kimang Khun**                                                    *kimang.khun@inria.fr*
*Univ. Grenoble Alpes, Inria, CNRS, Grenoble INP\*, LIG*
*38000 Grenoble, France*
*\*Institute of Engineering Univ. Grenoble Alpes*

**Reviewed on OpenReview:** *https://openreview.net/forum?id=Sh3RF9JowK&noteId=xrsTB4Cenz*

## Abstract

In this paper, we study the scalability of model-based algorithms learning the optimal policy of a discounted rested Markovian bandit problem with $n$ arms. There are two categories of model-based reinforcement learning algorithms: Bayesian algorithms (like PSRL), and optimistic algorithms (like UCRL2 or UCBVI). A naive application of these algorithms is not scalable because the state-space is exponential in $n$. In this paper, we construct variants of these algorithms specially tailored to Markovian bandits (MB) that we call MB-PSRL, MB-UCRL2, and MB-UCBVI. We consider an episodic setting with geometrically distributed episode length, and measure the performance of the algorithm in terms of regret (Bayesian regret for MB-PSRL and expected regret for MB-UCRL2 and MB-UCBVI). We prove that, for this setting, all algorithms have a low regret in $\tilde{O}(S\sqrt{nK})$ – where $K$ is the number of episodes, $n$ is the number of arms and $S$ is the number of states of each arm. Up to a factor $\sqrt{S}$, these regrets match the Bayesian minimax regret lower bound of $\Omega(\sqrt{SnK})$ that we also derive.

Even if their theoretical regrets are comparable, the *time complexities* of these algorithms vary greatly: We show that MB-UCRL2, as well as all algorithms that use bonuses on transition matrices have a time complexity that grows exponentially in $n$. In contrast, MB-UCBVI does not use bonuses on transition matrices and we show that it can be implemented efficiently, with a time complexity linear in $n$. Our numerical experiments show, however, that its empirical regret is large. Our Bayesian algorithm, MB-PSRL, enjoys the best of both worlds: its running time is linear in the number of arms and its empirical regret is the smallest of all algorithms. This is a new addition in the understanding of the power of Bayesian algorithms, that can often be tailored to the structure of the problems to learn.

## 1 Introduction

Markov decision processes (MDPs) are a powerful model to solve stochastic optimization problems. They suffer, however, from what is called the *curse of dimensionality*, which basically says that the state size of a Markov process is exponential in the number of system components. This implies that the complexity of computing an optimal policy is, in general, exponential in the number of system components. The same holds for general purpose reinforcement learning algorithms: they all have a regret and a runtime exponential in the number of components, so they also suffer from the same curse. Very few MDPs are known to escape from this curse of dimensionality. One of the most famous examples is the Markovian bandit problem in which a decision maker faces $n$ Markov reward processes (the $n$ components, that we will call the $n$ arms in the rest of the paper), and must decide which arm to activate at each decision epoch. Markovian bandit is a well structured MDP; and an optimal policy for such a system can be computed in $O(n)$, by using the

Gittins indices (computed for each arm independently), and its value can be computed by using retirement values (see for example Whittle (1996)). In this paper, we investigate how reinforcement learning algorithms can exploit these two advantages.

We consider an episodic setting with geometrically distributed episode length, in which the optimal strategy for a decision maker who know all parameters of the system is to use Gittins index policy. We study a specialization of PSRL (Osband et al., 2013) to Markovian bandits, that we call Markovian bandit posterior sampling (MB-PSRL) that consists in using PSRL with a prior tailored to Markovian bandits. We show that the Bayesian regret of MB-PSRL is sub-linear in the number of episodes and arms. We also provide an expected regret guarantee for two optimistic algorithms that we call MB-UCRL2 and MB-UCBVI, and that are based respectively on UCRL2 (Jaksch et al., 2010) and UCBVI (Azar et al., 2017). They both use modified confidence bounds adapted to Markovian bandit problems. The upper bound for their regret is similar to the bound for MB-PSRL. This shows that in terms of regret, the Bayesian approach (MB-PSRL) and the optimistic approach (MB-UCRL2 and MB-UCBVI) scale well with the number of arms. We also provide a Bayesian minimax regret lower bound for any learning algorithm in rested Markovian bandit problems with the aforementioned setting, which shows that the regret bounds that we obtain for the three algorithms are close to optimal.

The situation is radically different when considering the processing time: the runtime of MB-PSRL is linear (in the number of arms), while the runtime of MB-UCRL2 is exponential. We show that this is not an artifact of our implementation of MB-UCRL2 by exhibiting a Markovian bandit problem for which being optimistic in each arm is not optimistic in the global MDP. This implies that UCRL2 and its variants (Bourel et al., 2020; Fruit et al., 2018; Talebi & Maillard, 2018; Filippi et al., 2010) cannot be adapted to have efficient runtime in Markovian bandit problems unless an oracle gives the optimal policy. We argue that this non-scalability of UCRL2 and its variants is not a limitation of the optimistic approach but comes from the fact that UCRL2 relies on extended value iteration (Jaksch et al., 2010) needed to deal with upper confidence bounds on the transition matrices. We show that MB-UCBVI, an optimistic algorithm that does not add bonus on transition probabilities and hence does not rely on extended value iteration, does not suffer from the same problem. Its regret is sub-linear in the number of episodes, and arms (although larger than the regret of both MB-PSRL and MB-UCRL2), and its runtime is linear in the number of arms. This allows us to conclude that, on the one hand, if a weakly coupled MDP or a factored MDP can be solved efficiently when all the parameters are known, then the Bayesian approach is efficient both in terms of learning and computation time. On the other hand, knowing how to solve a weakly coupled MDP or a factored MDP efficiently is not sufficient for all optimistic algorithms to be computationally efficient.

We also conduct a series of numerical experiments to compare the performance of MB-PSRL, MB-UCRL2 and MB-UCBVI. They confirm the good behavior of MB-PSRL, both in terms of regret and computational complexity. These numerical experiments also show that the empirical regret of MB-UCBVI is larger than the regret of MB-PSRL and MB-UCRL2, confirming the comparisons between the upper bounds derived in Theorem 1. All this makes MB-PSRL the better choice between the three learning algorithms.

**Related work**   Markovian bandits have been applied to many problems such as single-machine scheduling, choosing a job in ressource constraint problems as well as other industrial research problems. Many applications can be found in Puterman (2014, Section 3.6) and Gittins et al. (2011). Gittins (1979) shows that rested Markovian bandit can be solved linearly in the number of arms using Gittins index policy. Therefore, several papers are focused on the complexity of computing Gittins index (Chakravorty & Mahajan, 2014; Gast et al., 2022). In this paper, we focus on rested Markovian bandit problems with discount factor $\beta < 1$ where all reward functions and transition matrices are unknown. A possible approach to learn under these conditions is to ignore the problem structure and view the Markovian bandit problem as a generic MDP. There are two main families of generic reinforcement learning algorithms with regret guarantees. The first one uses the *optimism in face of uncertainty* (OFU) principle. OFU methods build a confidence set for the unknown MDP and compute an optimal policy of the "best" MDP in the confidence set, *e.g.*, Bourel et al. (2020); Zhang & Ji (2019); Talebi & Maillard (2018); Fruit et al. (2017); Azar et al. (2017); Bartlett & Tewari (2012); Jaksch et al. (2010). UCRL2 (Jaksch et al., 2010) is a well known OFU algorithm. The second family uses a Bayesian approach, the posterior sampling method introduced by Thompson (1933). Such algorithms

keep a posterior distribution over possible MDPs and execute the optimal policy of a sampled MDP, see *e.g.*, Ouyang et al. (2017); Agrawal & Jia (2017); Gopalan & Mannor (2015); Osband et al. (2013). PSRL (Osband et al., 2013) is a classical example of Bayesian learning algorithm. All these algorithms, based on OFU or on Bayesian principles, have sub-linear bounds on the regret, which means that they provably learn the optimal policy. Yet, applied as-is to Markovian bandit problems, these bounds grow exponentially with the number of arms.

Our work is not the first attempt to exploit the structure of a MDP to improve learning. Factored MDPs (the state space can be factored into $n \in \mathbb{N}^*$ components) are investigated in Guestrin et al. (2003), where asymptotic convergence to the optimal policy is proved to scale polynomially in the number of components. The regret of learning algorithms in factored MDP with a factored action space is considered by Tian et al. (2020); Rosenberg & Mansour (2020); Xu & Tewari (2020); Osband & Van Roy (2014). Our work differs substantially from these. First, the Markovian bandit problem is not a factored MDP because the action space is global and cannot be factored. Second, our reward is discounted over an infinite horizon while factored MDPs have been analyzed with no discount. Finally, and most importantly, the factored MDP framework assumes that the successive optimal policies are computed by an unspecified solver (oracle). There is no guarantee that the time complexity of this solver scales linearly with the number of components, especially for OFU-based algorithms. For Markovian bandits, we get an additional leverage: when all parameters are known, the Gittins index policy is known to be an optimal policy and its computational complexity is linear in the number of arms. This reveals an interesting difference between Bayesian and extended value based algorithms (the former being scalable and not the latter). This difference is not present in the literature about factored MDPs because such papers do not consider the time complexity.

Our Markovian bandit setting is known in the literature as *rested* or *restful* bandit or a *family of alternative bandit processes*. Tekin & Liu (2012) consider a non-discounted setting, $\beta = 1$, and provide algorithms with logarithmic regret guarantee for *rested* as well as *restless* settings (a generalization of rested). However, they consider a notion of regret known as *weak regret* that measures how fast the learning algorithm identifies the best arm in stationary regime. So, it ignores the learning behavior at the beginning of learning process. In contrast, we consider the discounted rested bandit setting in which the regret of Tekin & Liu (2012) makes no more senses due to the discount factor and we propose a regret definition that is frequently used in reinforcement learning literature and captures the performance of a learning algorithm during the whole learning process. In addition, Ortner et al. (2012); Jung & Tewari (2019); Wang et al. (2020b) consider a non-discounted restless bandit setting in which only the state of chosen arms are observed by the learner. Ortner et al. (2012); Wang et al. (2020b) propose optimistic algorithms for infinite-horizon setting and provide regret bounds that are sub-linear in time. Again the discounted case is not considered in these papers while it is particularly interesting because learning algorithms can leverage the optimal Gittins index policy. Jung & Tewari (2019) propose a Bayesian algorithm in the episodic finite-horizon setting and also provide a regret bound that is sub-linear in the number of episodes. However, the computational complexity is not studied in their work (the algorithm of Ortner et al. (2012) is intractable while the ones of Jung & Tewari (2019); Wang et al. (2020b) rely on the unspecified problem solver called *oracle*). Contrarily, we provide both performance guarantee and computational complexity analysis of each algorithm that we consider in this paper. Finally, Killian et al. (2021) consider a more general setting of restless bandits in which each arm is itself a MDP and the learner has to decide which arms to choose and which action to execute on each chosen arm under a global action constraint. The authors propose a Lagrangian suboptimal policy to solve the restless bandit problem with known parameters and a sampling algorithm to learn their Lagrangian policy when the parameters are unknown. Unfortunately, no performance guarantee is provided in their work.

Since index policies scale with the number of arms, using Q-learning approaches to learn such a policy is also popular, see *e.g.*, Avrachenkov & Borkar (2022); Fu et al. (2019); Duff (1995). Duff (1995) addresses the same Markovian bandit problem as we do: their algorithm learns the optimal value in the restart-in-state MDP (Katehakis & Veinott Jr, 1987) for each arm and uses Softmax exploration to solve the exploration-exploitation dilemma. As mentioned on page 250 of Auer et al. (2002), however, there exists no finite-time regret bounds for this algorithm. Furthermore, tuning its hyperparameters (learning rate and temperature) is rather delicate and unstable in practice.

## 2 Markovian bandit problem

In this section, we introduce the Markovian bandit problem and recall the notion of Gittins index when the parameters $(\boldsymbol{r}^a, Q^a)$ of all arms are known.

### 2.1 Definitions and main notations

We consider a Markovian bandit problem with $n$ arms. Each arm $\langle \mathcal{S}^a, \boldsymbol{r}^a, Q^a \rangle$ for $a \in \{1, \dots, n\} =: [n]$ is a Markov reward process with a finite state space $\mathcal{S}^a$ of size $S$. Each arm has a mean reward vector, $\boldsymbol{r}^a \in [0,1]^S$, and a transition matrix $Q^a$. When Arm $a$ is activated in state $x_a \in \mathcal{S}^a$, it moves to state $y_a \in \mathcal{S}^a$ with probability $Q^a(x_a, y_a)$. This provides a reward whose expected value is $r^a(x_a)$. Without loss of generality, we assume that the state spaces of the arms are pairwise distinct: $\mathcal{S}^a \cap \mathcal{S}^b = \emptyset$ for $a \neq b$. In the following, the state of an arm $a$ will always be denoted with an index $a$: we will denote such a state by $x_a$ or $y_a$. As state spaces are disjoint, this allows us to simplify the notation by dropping the index $a$ from the reward and transition matrix: when convenient, we will denote them by $r(x_a)$ instead of $r^a(x_a)$ and by $Q(x_a, y_a)$ instead of $Q^a(x_a, y_a)$ since no confusion is possible.

At time 1, the global state $\boldsymbol{X}_1$ is distributed according to some initial distribution $\rho$ over the global state space $\mathcal{E} = \mathcal{S}^1 \times \dots \times \mathcal{S}^n$. At time $t$, the decision maker observes the states[1] of all arms, $\boldsymbol{X}_t = (X_{t,1} \ \dots \ X_{t,n})$, and chooses which arm $A_t$ to activate. This problem can be cast as a MDP – that we denote by $M$ – with state space $\mathcal{E}$ and action space $[n]$. Let $a \in [n]$ and $\boldsymbol{x}, \boldsymbol{y} \in \mathcal{E}$. If the state at time $t$ is $\boldsymbol{X}_t = \boldsymbol{x}$, the chosen arm is $A_t = a$, then the agent receives a random reward $R_t$ drawn from some distribution on $[0,1]$ with mean $r(x_a)$ and the MDP $M$ transitions to state $\boldsymbol{X}_{t+1} = \boldsymbol{y}$ with probability $P^a(\boldsymbol{x}, \boldsymbol{y})$ that satisfies:

$$P^a(\boldsymbol{x}, \boldsymbol{y}) = \begin{cases} Q(x_a, y_a) & \text{if } x_{a'} = y_{a'} \text{ for all } a' \neq a; \\ 0 & \text{otherwise.} \end{cases} \tag{1}$$

That is, the active arm makes a transition while the other arms remain in the same state.

Let $\Pi$ be the set of deterministic policies, *i.e.*, the set of functions $\pi : \mathcal{E} \mapsto [n]$. For the MDP $M$, we denote by $V_M^\pi(\boldsymbol{x})$ the expected cumulative discounted reward of $M$ under policy $\pi$ starting from an initial state $\boldsymbol{x}$:

$$V_M^\pi(\boldsymbol{x}) = \mathbb{E}\left[ \sum_{t=1}^\infty \beta^{t-1} R_t \mid \boldsymbol{X}_1 = \boldsymbol{x}, A_t = \pi(\boldsymbol{X}_t) \right].$$

An alternative definition of $V$ is to consider a finite-horizon problem with a geometrically distributed length. Indeed, let $H$ be a time-horizon geometrically distributed with parameter $1 - \beta > 0$. We have

$$V_M^\pi(\boldsymbol{x}) = \mathbb{E}\left[ \sum_{t=1}^H R_t \mid \boldsymbol{X}_1 = \boldsymbol{x}, A_t = \pi(\boldsymbol{X}_t) \right]. \tag{2}$$

**Problem 1.** *Given a Markovian bandit $M$ with $n$ arms, each is a Markov reward process $\langle \mathcal{S}^a, \boldsymbol{r}^a, Q^a \rangle$ with a finite state space of size $S$, find a policy $\pi : \mathcal{S}^1 \times \dots \times \mathcal{S}^n \mapsto [n]$ that maximizes $V_M^\pi(\boldsymbol{x})$ for any state $\boldsymbol{x}$ distributed according to initial global state distribution $\rho$.*

By a small abuse of notation, we denote by $V_M^\pi(\rho)$ the expected reward when the initial state is randomly generated according to $\rho : V_M^\pi(\rho) = \sum_{\boldsymbol{x}} \rho(\boldsymbol{x}) V_M^\pi(\boldsymbol{x})$. A policy $\pi_*$ is optimal for Problem 1 if $V_M^{\pi_*}(\boldsymbol{x}) \geq V_M^\pi(\boldsymbol{x})$ for all $\pi \in \Pi$ and $\boldsymbol{x} \in \mathcal{E}$. By Puterman (2014), such a policy exists and does not depend on $\boldsymbol{x}$ (or $\rho$). One well-known optimal policy is the Gittins index policy, defined below.

---

[1]Throughout the paper, we use capital letters (like $X_t$) to denote random variables and small letter (like $\boldsymbol{x}$) to denote their realizations. Bold letters ($\boldsymbol{X}_t$ or $\boldsymbol{x}$) design vectors. Normal letters ($X_{t,a}$ or $x_a$) are for scalar values.

## 2.2 Gittins index policy

It is possible to compute an optimal policy $\pi_*$ for Problem 1 in a reasonable amount of time using the so called Gittins indices: Gittins (1979) defines the *Gittins index* for any arm $a$ in state $x_a \in \mathcal{S}^a$ as

$$\text{GIndex}(x_a) = \sup_{\tau > 0} \frac{\mathbb{E}\left[\sum_{t=1}^{\tau} \beta^{t-1} r^a(Z_t) \mid Z_1 = x_a\right]}{\mathbb{E}\left[\sum_{t=1}^{\tau} \beta^{t-1} \mid Z_1 = x_a\right]}, \tag{3}$$

where $Z$ is a Markov chain whose transitions are given by $Q^a$ and $\tau$ can be any stopping time adapted to the natural filtration of $(Z_t)_{t \geq 1}$. So, Gittins index can be considered as the maximal reward density over time of an arm at the given state.

Gittins (1979) shows that activating the arm having the largest current index is an optimal policy. Such a policy can be computed very efficiently: The computation of the indices of an arm with $S$ states can be done in $O(S^3)$ arithmetic operations, which means that the computation of Gittins index policy is linear in the number of arms as it takes $O(nS^3)$ arithmetic operations. For more details about Gittins indices and optimality, we refer to Gittins et al. (2011); Weber (1992). For a survey on how to compute Gittins indices, we refer to Chakravorty & Mahajan (2014), and to Gast et al. (2022) for a recent paper that shows how to compute Gittins index in subcubic time (*i.e.*, $o(S^3)$) for each of the $n$ arms).

## 3 Online learning and episodic regret

We now consider an extension of Problem 1 in which the decision maker does not know the transition matrices nor the rewards. Our goal is to design a reinforcement learning algorithm that learns the optimal policy from past observations. Similarly to what is done for finite-horizon reinforcement learning with deterministic horizon – see *e.g.*, Zanette & Brunskill (2019); Jin et al. (2018); Azar et al. (2017); Osband et al. (2013) – we consider a decision maker that faces a sequence of independent replicas of the same Markovian bandit problem, where the transitions and the rewards are drawn independently for each episode. What is new here is that the time horizon $H$ is random and has a geometric distribution with expected value $1/(1 - \beta)$. It is drawn independently for each episode. This implies that Gittins index policy is optimal for a decision maker that would know the transition matrices and rewards.

In this paper, we consider *episodic learning algorithms*. Let $H_1, \ldots, H_k$ be the sequence of random episode lengths and let $t_k := 1 + \sum_{i=1}^{k-1} H_i$ be the starting time of the $k$th episode. Let $\mathcal{O}_{k-1} := \{\boldsymbol{X}_1, A_1, R_1, \ldots, \boldsymbol{X}_{t_k-1}, A_{t_k-1}, R_{t_k-1}\}$ denote the observations made prior and up to episode $k$. An *Episodic Learning Algorithm* $\mathcal{L}$ is a function that maps observations $\mathcal{O}_{k-1}$ to $\mathcal{L}(\mathcal{O}_{k-1})$, a probability distribution whose support is $\Pi$. At the beginning of episode $k$, the algorithm samples $\pi_k \sim \mathcal{L}(\mathcal{O}_{k-1})$ and uses this policy during the whole $k$th episode. Note that one could also design algorithms where learning takes place inside each episode. We will see later that episodic learning as described here is enough to design algorithms that are essentially optimal, in the sense given by Theorem 1 and Theorem 2.

For an instance $M$ of a Markovian bandit problem and a total number of episodes $K$, we denote by $\text{Reg}(K, \mathcal{L}, M)$ the regret of a learning algorithm $\mathcal{L}$, defined as

$$\text{Reg}(K, \mathcal{L}, M) := \sum_{k=1}^{K} V_M^{\pi_*}(\boldsymbol{X}_{t_k}) - V_M^{\pi_k}(\boldsymbol{X}_{t_k}). \tag{4}$$

It is the sum over all episodes of the value of the optimal policy $\pi_*$ minus the value obtained by applying the policy $\pi_k$ chosen by the algorithm for episode $k$. In what follows, we will provide bounds on the expected regret.

A no-regret algorithm is an algorithm $\mathcal{L}$ such that its expected regret $\mathbb{E}\left[\text{Reg}(K, \mathcal{L}, M)\right]$ grows sub-linearly in the number of episodes $K$. This implies that the expected regret over episode $k$ converges to 0 as $k$ goes to infinity. Such an algorithm learns an optimal policy of Problem 1.

Note that, for discounted MDPs, an alternative regret definition (used for instance by He et al. (2021)) is to use the non-episodic version $\sum_{t=1}^{T}(V_M^{\pi_*}(\boldsymbol{X}_t) - V_M^{\pi_t}(\boldsymbol{X}_t))$. In our definition at Equation 4, we use an episodic

approach where the process is restarted according to $\rho$ after each episode of geometrically distributed length $H_k$.

# 4 Learning algorithms for Markovian bandits

In what follows, we present three algorithms having a regret that grows like $\tilde{O}(S\sqrt{nK})$, that we call MB-PSRL, MB-UCRL2 and MB-UCBVI. As their names suggest, these algorithms are adaptation of PSRL, UCRL2 and UCBVI to Markovian bandit problems that intend to overcome the exponentiality in $n$ of their regret. The structure of the three MB-* algorithms are similar and is represented in Algorithm 1. All algorithms are episodic learning algorithms. At the beginning of each episode, a MB-* learning algorithm computes a new policy $\pi_k$ that will be used during an episode of geometrically distributed length. The difference between the three algorithms lies in the way this new policy $\pi_k$ is computed. MB-PSRL uses posterior sampling while MB-UCRL2 and MB-UCBVI use optimism. We detail the three algorithms below.

---

**Algorithm 1** Pseudo-code of the three MB-* algorithms.

---

**input** Discount factor $\beta$, initial distribution $\rho$ (and a prior distribution $\{\phi^a\}_{a\in[n]}$ for MB-PSRL)
1: **for** episodes $k = 1, 2, \dots$ **do**
2:    Compute a new policy $\pi_k$ (using posterior sampling or optimism).
3:    Set $t_k \leftarrow 1 + \sum_{i=1}^{k-1} H_i$, sample $\boldsymbol{X}_{t_k} \sim \rho$ and $H_k \sim \text{Geom}(1-\beta)$.
4:    **for** $t \leftarrow t_k$ **to** $t_k + H_k - 1$ **do**
5:       Activate arm $A_t = \pi_k(\boldsymbol{X}_t)$.
6:       Observe $R_t$ and $\boldsymbol{X}_{t+1}$.
7:    **end for**
8: **end for**

---

## 4.1 MB-PSRL

MB-PSRL starts with a prior distribution $\phi^a$ over the parameters $(\boldsymbol{r}^a, Q^a)$. At the start of each episode $k$, MB-PSRL computes a posterior distribution of parameters $\phi^a(\cdot \mid \mathcal{O}_{k-1})$ for each arm $a \in [n]$ and samples parameters $(\boldsymbol{r}_k^a, Q_k^a)$ from $\phi^a(\cdot \mid \mathcal{O}_{k-1})$ for each arm. Then, MB-PSRL uses $\{(\boldsymbol{r}_k^a, Q_k^a)\}_{a\in[n]}$ to compute the Gittins index policy $\pi_k$ that is optimal for the sampled problem. The policy $\pi_k$ is then used for the whole episode $k$. Note that as $\pi_k$ is a Gittins index policy, it can be computed efficiently.

The difference between PSRL and MB-PSRL is mostly that MB-PSRL uses a prior distribution tailored to Markovian bandit. The only hyperparameter of MB-PSRL is the prior distribution $\phi$. As we see in Appendix E, MB-PSRL seems robust to the choice of the prior distribution, even if a coherent prior gives a better performance than a misspecified prior, similarly to what happens for Thompson's sampling (Russo et al., 2018).

## 4.2 MB-UCRL2

At the beginning of each episode $k$, MB-UCRL2 computes the following quantities for each state $x_a \in \mathcal{S}^a$: $N_{k-1}(x_a)$ the number of times that Arm $a$ is activated before episode $k$ while being in state $x_a$, and $\hat{r}_{k-1}(x_a)$, and $\hat{Q}_{k-1}(x_a, \cdot)$ are the empirical means of $r(x_a)$ and $Q(x_a, \cdot)$. We define the confidence bonuses $b_{k-1}^r(x_a) := \sqrt{\frac{\log(2SnKt_k)}{2\max\{1, N_{k-1}(x_a)\}}}$ and $b_{k-1}^Q(x_a) := \sqrt{\frac{2\log(SnK2^S t_k)}{\max\{1, N_{k-1}(x_a)\}}}$. This defines a confidence set $\mathbb{M}_k$ as follows: a Markovian bandit problem $M'$ is in $\mathbb{M}_k$ if for all $a \in [n]$ and $x_a \in \mathcal{S}^a$:

$$|r'(x_a) - \hat{r}_{k-1}(x_a)| \leq b_{k-1}^r(x_a) \text{ and } \|Q'(x_a, \cdot) - \hat{Q}_{k-1}(x_a, \cdot)\|_1 \leq b_{k-1}^Q(x_a). \tag{5}$$

MB-UCRL2 then chooses a policy $\pi_k$ that is optimal for the most optimistic problem $M_k \in \mathbb{M}_k$:

$$\pi_k \in \arg\max_\pi \max_{M' \in \mathbb{M}_k} V_{M'}^\pi(\rho). \tag{6}$$

Note that as we explain later in Section 6.1, we believe that there is no efficient algorithm to compute the best optimistic policy $\pi_k$ of Equation 6.

Compared to a vanilla implementation of UCRL2, MB-UCRL2 uses the structure of the Markovian bandit problem: The constraints Equation 5 are on $Q$ whereas vanilla UCRL2 uses constraints on the full matrix $P$ (defined in Equation 1). This leads MB-UCRL2 to use the bonus term that scales as $\sqrt{S/N_{k-1}(x_a)}$ whereas vanilla UCRL2 would use the term in $\sqrt{S^n/N_{k-1}(\boldsymbol{x}, a)}$.

### 4.3 MB-UCBVI

At the beginning of episode $k$, MB-UCBVI uses the same quantities $N_{k-1}(x_a)$, $\hat{r}_{k-1}(x_a)$, and $\hat{Q}_{k-1}(x_a, \cdot)$ as MB-UCRL2. The difference lies in the definition of the bonus terms. While MB-UCRL2 uses a bonus on the reward and on the transition matrices, MB-UCBVI defines a bonus $b_{k-1}^a(x_a) := \frac{1}{1-\beta} \sqrt{\frac{\log(2SnKt_k)}{2\max\{1, N_{k-1}(x_a)\}}}$ that is used on the reward only. MB-UCBVI computes the Gittins index policy $\pi_k$ that is optimal for the bandit problem $\{(\hat{\boldsymbol{r}}_{k-1}^a + b_{k-1}^a, \hat{Q}_{k-1}^a)\}_{a \in [n]}$.

Similarly to the case of UCRL2, a vanilla implementation of UCBVI would use a bonus that scales exponentially with the number of arms. MB-UCBVI makes an even better use of the structure of the learned problem because the optimistic MDP $\{(\hat{\boldsymbol{r}}_{k-1}^a + b_{k-1}^a, \hat{Q}_{k-1}^a)\}_{a \in [n]}$ is still a Markovian bandit problem. This implies that the optimistic policy $\pi_k$ is a Gittins index policy, and that can therefore be computed efficiently.

## 5 Regret analysis

In this section, we first present upper bounds on the expected regret of the three learning algorithms. These bounds are sub-linear in the number of episodes (hence the three algorithms are no-regret algorithms) and sub-linear in the number of arms. We then derive a minimax lower bound on the Bayesian regret of any learning algorithm in the Markovian bandit problem.

### 5.1 Upper bounds on regret

The theorem below provides upper bounds on the regret of the three algorithms presented in Section 4. Note that since MB-PSRL is a Bayesian algorithm, we consider its *Bayesian regret*, that is the expectation over all possible models. More precisely, if the unknown MDP $M$ is drawn from a prior distribution $\phi$, the *Bayesian regret* of a learning algorithm $\mathcal{L}$ is $\text{BayReg}(K, \mathcal{L}, \phi) = \mathbb{E}[\text{Reg}(K, \mathcal{L}, M)]$, where the expectation is taken over all possible values of $M \sim \phi$ and all possible runs of the algorithm. The expected regret $\mathbb{E}[\text{Reg}(K, \mathcal{L}, M)]$ is defined by taking the expectation over all possible runs of the algorithm.

**Theorem 1.** *Let* $f(S, n, K, \beta) = Sn\left(\log K/(1-\beta)\right)^2 + \sqrt{SnK}\left(\log K/(1-\beta)\right)^{3/2}$. *There exists universal constants* $C, C'$ *and* $C''$ *independent of the model (i.e., that do not depend on $S$, $n$, $K$ and $\beta$) such that:*

- *For any prior distribution $\phi$:*

$$\text{BayReg}(K, \textit{MB-PSRL}, \phi) \leq C\left(\sqrt{S} + \log \frac{SnK \log K}{1-\beta}\right) f(S, n, K, \beta),$$

- *For any Markovian bandit model $M$:*

$$\mathbb{E}[\text{Reg}(K, \textit{MB-UCRL2}, M)] \leq C'\left(\sqrt{S} + \log \frac{SnK \log K}{1-\beta}\right) f(S, n, K, \beta),$$

$$\mathbb{E}[\text{Reg}(K, \textit{MB-UCBVI}, M)] \leq C''\left(\frac{\sqrt{S}}{1-\beta}\right)\left(\log \frac{SnK \log K}{1-\beta}\right) f(S, n, K, \beta),$$

We provide a sketch of proof below. The detailed proof is provided in Appendix A in the supplementary material.

This theorem calls for several comments. First, it shows that when $K \geq Sn/(1-\beta)$, the regret of MB-PSRL and MB-UCRL2 is smaller than

$$\tilde{O}\left(\frac{S\sqrt{nK}}{(1-\beta)^{3/2}}\right), \tag{7}$$

where the notation $\tilde{O}$ means that all logarithmic terms are removed. The regret of MB-UCBVI has an extra $1/(1-\beta)$ factor.

Hence, the regret of the three algorithms is sub-linear in the number of episodes $K$ which means that they all are no-regret algorithms. This regret bound is sub-linear in the number of arms which is very significant in practice when facing a large number of arms. Note that directly applying PSRL, UCRL2 or UCBVI would lead to a regret in $\tilde{O}\left(S^n\sqrt{nK}\right)$ or $\tilde{O}\left(\sqrt{nS^nK}\right)$, which is exponential in $n$.

Second, the upper bound on the expected regret of MB-UCRL2 (and of MB-UCBVI) is a guarantee for a specific problem $M$ while the bound on Bayesian regret of MB-PSRL is a guarantee in average overall the problems drawn from the prior $\phi$. Hence, the bounds of MB-UCRL2 and MB-UCBVI are stronger guarantee compared to the one of MB-PSRL. Yet, as we will see later in the numerical experiments reported in Section 7, MB-PSRL seems to have a smaller regret in practice, even when the problem does not follow the correct prior. An interesting open question would be to find a prior that would guarantee that MB-PSRL has a good worst-case regret bound. We do not know if such a prior exists and to the best of our knowledge, this question is also open for the classical PSRL. Note that there exist Bayesian type algorithms with worst-case guarantees, see *e.g.*, (Ishfaq et al., 2021; Agrawal et al., 2021; Wang et al., 2020a; Agrawal & Jia, 2017) but they contain an optimistic part and it is not clear how to implement them in an efficient manner for Markovian bandits.

Third, the result of Theorem 1 is the statistical evaluation of the three learning algorithms and does not require them to use Gittins index policy (in particular, MB-UCRL2 does not use Gittins index policy). What is required is that policy $\pi_k$ is optimal for the sampled problem $M_k$ for MB-PSRL (so that Lemma 5 applies) or for the optimistic problem $M_k$ for MB-UCBVI (so that (11) is valid). Indeed, instead of using Gittins index policy for MB-PSRL or MB-UCBVI, assume that we have access to an oracle that provides an optimal policy for any given Markovian bandit problem. Then, the upper bound on regret in Theorem 1 still holds when MB-PSRL and MB-UCBVI use the oracle to compute policy $\pi_k$. Gittins index policy is required only for the runtime evaluation as we will see in Section 6.

Finally, our bound in Equation 7 is linear in $S$, the state space size of each arm. Having a regret bound linear in the state space size is currently state-of-the-art for Bayesian algorithms, see *e.g.*, Agrawal & Jia (2017); Ouyang et al. (2017) and our discussion in Appendix A.3.4. For optimistic algorithms, the best regret bounds are linear in the square root of the state size because they use Bernstein's concentration bounds instead of Weissman's inequality (Azar et al., 2017), yet this approach does not work in our setting due to the randomness of episode's length and the bound of MB-UCBVI depends linearly on $S$. We discuss more about this in Appendix A.5.4. UCBVI has also been studied in the discounted case by He et al. (2021). They use, however, a different definition of regret, making their bound on the regret hardly comparable to ours.

**Sketch of proof**

A crucial ingredient of our proof is to work with the value function over a random finite time horizon ($W$ defined below), instead of working directly with the discounted value function $V$. For a given model $M$, and a deterministic policy $\pi$, a horizon $H$ and a time step $h \leq H$, we define by $W^{\pi}_{M,h:H}(\boldsymbol{x})$ the value function of policy $\pi$ over the finite time horizon $H - h + 1$ when starting in $\boldsymbol{x}$ at time $h$. It is defined as

$$W^{\pi}_{M,h:H}(\boldsymbol{x}) := r^{\pi}(\boldsymbol{x}) + \sum_{\boldsymbol{y} \in \mathcal{E}} P^{\pi}(\boldsymbol{x}, \boldsymbol{y}) W^{\pi}_{M,h+1:H}(\boldsymbol{y}), \tag{8}$$

with $W^{\pi}_{M,H:H}(\boldsymbol{x}) := r^{\pi}(\boldsymbol{x})$ and where $r^{\pi}$ and $P^{\pi}$ are reward vector and state transition matrix when following policy $\pi$.

By definitions of $W$ in Equation 8 and $V$ in Equation 2, for a fixed model $M$, a policy $\pi$ and a state $\boldsymbol{x}$, and a time horizon $H$ that is geometrically distributed, one has $V_M^\pi(\boldsymbol{x}) = \mathbb{E}\left[W_{M,1:H}^\pi(\boldsymbol{x})\right]$.

This characterization is important in our proof. Since the episode length $H_k$ is independent of the observations available before episode $k$, $\mathcal{O}_{k-1}$, for any policy $\pi_k$ that is independent of $H_k$, one has

$$\mathbb{E}\left[V_M^{\pi_k}(\boldsymbol{X}_{t_k}) \mid \mathcal{O}_{k-1}, \pi_k\right] = \mathbb{E}\left[W_{M,1:H_k}^{\pi_k}(\boldsymbol{X}_{t_k}) \mid \mathcal{O}_{k-1}, \pi_k\right]. \tag{9}$$

In the above Equation 9, the expectation is taken over all initial state $\boldsymbol{X}_{t_k}$ and all possible horizon $H_k$.

Equation 9 will be very useful in our analysis as it allows us to work with either $V$ or $W$ interchangeably. While the proof of MB-PSRL could be done by only studying the function $W$, the proof of MB-UCRL2 and MB-UCBVI will use the expression of the regret as a function of $V$ to deal with the non-determinism. Indeed, at episode $k$, all algorithms compare the optimal policy $\pi_*$ (that is optimal for the true MDP $M$) and a policy $\pi_k$ chosen by the algorithm (that is optimal for a MDP $M_k$ that is either sampled by MB-PSRL or chosen by an optimistic principle). The quantity $\Delta_k := W_{M,1:H_k}^{\pi_*}(\boldsymbol{X}_{t_k}) - W_{M,1:H_k}^{\pi_k}(\boldsymbol{X}_{t_k})$ equals:

$$\underbrace{W_{M,1:H_k}^{\pi_*}(\boldsymbol{X}_{t_k}) - W_{M_k,1:H_k}^{\pi_k}(\boldsymbol{X}_{t_k})}_{(A)} + \underbrace{W_{M_k,1:H_k}^{\pi_k}(\boldsymbol{X}_{t_k}) - W_{M,1:H_k}^{\pi_k}(\boldsymbol{X}_{t_k})}_{(B)}. \tag{10}$$

The analysis of the term (B) is similar for the three algorithms: it is bounded by the distance between the sampled MDP $M_k$ and the true MDP $M$ that can in turn be bounded by using a concentration argument (Lemma 1) based on Hoeffding's and Weissman's inequalities. Compared with the literature (Azar et al., 2017; Ouyang et al., 2017), our proof leverages on taking conditional expectations, making all terms whose conditional expectation is zero disappear. One of the main technical hurdle is to deal with the random episodes lengths $H_1, \ldots, H_k$. This is required in our approach and is not needed in the classical analysis of finite horizons problems.

The analysis of (A) depends heavily on the algorithm used. The easiest case is PSRL: As our setting is Bayesian, the expectation of the first term (A) with respect to the model is zero (see Lemma 5). The case of MB-UCRL2 and MB-UCBVI are harder. In fact, our bonus terms are specially designed so that $V_{M_k}^{\pi_k}(\boldsymbol{x})$ is an optimistic upper bound of the true value function with high probability, that is:

$$V_{M_k}^{\pi_k}(\boldsymbol{x}) = \max_\pi \max_{M' \in \mathbb{M}_k} V_{M'}^\pi(\boldsymbol{x}) \geq V_M^{\pi_*}(\boldsymbol{x}). \tag{11}$$

This requires the use of $V$ and not $W$ and it is used to show that the expectation of the term (A) of Equation 10 cannot be positive.

## 5.2 Bayesian minimax lower bound

After obtaining upper bounds on the regret, a natural question is: can we do better? Or in other terms, does there exist a learning algorithm with a smaller regret? To answer this question, the metric used in the literature is the notion of minimax lower bound: for a given set of parameters $(S, n, K, \beta)$, a minimax lower bound is a lower bound on the quantity $\inf_\mathcal{L} \sup_M \text{Reg}(K, \mathcal{L}, M)$, where the supremum is taken among all possible models that have parameters $(S, n, K, \beta)$ and the infimum is taken over all possible learning algorithms. The next theorem provides a lower bound on the Bayesian regret. It is therefore stronger than a minimax bound for two reasons: First, the Bayesian regret is an average over models, which means that there exists at least one model that has a larger regret than the Bayesian lower bound; And second, in Theorem 2, we allow the algorithm to depend on the prior distribution $\phi$ and to use this information.

**Theorem 2** (Lower bound). *For any state size $S$, number of arms $n$, discount factor $\beta$ and number of episodes $K \geq 16S$, there exists a prior distribution $\phi$ on Markovian bandit problems with parameters $(S, n, K, \beta)$ such that, for any learning algorithm $\mathcal{L}$:*

$$\text{BayReg}(K, \mathcal{L}, \phi) \geq \frac{1}{60}\sqrt{\frac{SnK}{(1-\beta)}}. \tag{12}$$

The proof is given in Appendix B and uses a counterexample inspired by the one of Jaksch et al. (2010). Note that for general MDPs, the minimax lower bound obtained by Osband & Van Roy (2016); Jaksch et al. (2010) says that a learning algorithm cannot have a regret smaller than $\Omega\big(\sqrt{\tilde{S}\tilde{A}\tilde{T}}\big)$, where $\tilde{S}$ is the number of states of the MDP, $\tilde{A}$ is the number of actions and $\tilde{T}$ is the number of time steps. Yet, the lower bound of Osband & Van Roy (2016); Jaksch et al. (2010) is not directly applicable to our case with $\tilde{S} = S^n$ because Markovian bandit problems are very specific instances of MDPs and this can be exploited by the learning algorithm. Also note that this lower bound on the Bayesian regret is also a lower bound on the expected regret of any non-Bayesian algorithm for any MDP model $M$.

Apart from the logarithmic terms, the lower bound provided by Theorem 2 differs from the bound of Theorem 1 by a factor $\sqrt{S}/(1-\beta)$. This factor is similar to the one observed for PSRL and UCRL2 (Osband et al., 2013; Jaksch et al., 2010). There are various factors that could explain this. We believe that the extra factor $1/(1-\beta)$ might be half due to the episodic nature of MB-PSRL and MB-UCRL2 (when $1/(1-\beta)$ is large, algorithms with internal episodic updates might have smaller regret) and half due to the fact that the lower bound of Theorem 2 is not optimal and could include a term $1/\sqrt{1-\beta}$ (similar to the term $O(\sqrt{D})$ of the lower bound of Osband & Van Roy (2016); Jaksch et al. (2010)). The factor $\sqrt{S}$ between our two bounds comes from our use of Weissman's inequality. It might be possible that our regret bounds are not optimal with respect to this term although such an improvement cannot be obtained using the same approach of Azar et al. (2017).

# 6 Scalability of learning algorithms for Markovian bandits

Historically, Problem 1 was considered unresolved until Gittins (1979) proposed Gittins indices. This is because previous solutions were based on Dynamic Programming in the global MDP which are computationally expensive. Hence, after establishing regret guarantees, we are now interested in the computational complexity of our learning algorithms, which is often disregarded in the learning literature.

## 6.1 MB-PSRL and MB-UCBVI are scalable

If one excludes the simulation of the MDP, the computational cost of MB-PSRL and MB-UCBVI of each episode is low. For MB-PSRL, its cost is essentially due to three components: Updating the observations, sampling from the posterior distribution and computing the optimal policy. The first two are relatively fast when the conjugate posterior has a closed form: updating the observation takes $O(1)$ at each time, and sampling from the posterior can be done in $O(nS^2)$ – more details on posterior distributions are given in Appendix D. When the conjugate posterior is implicit (*i.e.*, under the integral form), the computation can be higher but remains linear in the number of arms. For MB-UCBVI, the cost is due to two components: computing the bonus terms and computing the Gittins policy for the optimistic MDP. Computing the bonus is linear in the number of bandits and the length of the episode. As explained in Section 2.2, the computation of the Gittins index policy for a given problem can be done in $O(nS^3)$. Hence, MB-PSRL and MB-UCBVI have a regret and a runtime both linear in the number of arms.

## 6.2 MB-UCRL2 is not scalable because it cannot use an Index Policy

While MB-UCRL2 has a regret equivalent to the one of MB-PSRL, its computational complexity, and in particular the complexity of computing an *optimistic* policy that maximizes Equation 6 does not scale with $n$. Such a policy can be computed by using *extended value iteration* (Jaksch et al., 2010). This computation is polynomial in the number of states of the global MDP and is therefore exponential in the number of arms, precisely $O(nS^{2n})$. For MB-PSRL (or MB-UCBVI), the computation is easier because the sampled (optimistic) MDP is a Markovian bandit problem. Hence, using Gittins Theorem, computing the optimal policy can be done by computing local indices. In the following, we show that it is not possible to solve Equation 6 by using local indices. This suggests that MB-UCRL2 (nor any of the modifications of UCRL2's variants that would use extended value iteration) cannot be implemented efficiently.

More precisely, to find an optimistic policy (that satisfies Equation 11), UCRL2 and its variants, *e.g.*, KL-UCRL (Filippi et al., 2010), compute a policy $\pi_k$ that is optimal for the most optimistic MDP in $\mathbb{M}_k$. This

can be done by using extended value iteration. We now show that this cannot be replaced by the computation of local indices.

Let us consider that the estimates and confidence bounds for a given arm $a$ are $\hat{\mathcal{B}}^a := (\hat{\boldsymbol{r}}^a, \hat{Q}^a, b_a^r, b_a^Q)$. We say that an algorithm computes indices locally for Arm $a$ if for each $x_a \in \mathcal{S}^a$, it computes an index $I^{\hat{\mathcal{B}}^a}(x_a)$ by using only $\hat{\mathcal{B}}^a$ but not $\hat{\mathcal{B}}^{a'}$ for any $a' \neq a$. We denote by $\pi^{I(\hat{\mathcal{B}})}$ the index policy that uses index $I^{\hat{\mathcal{B}}^a}$ for arm $a$ and by $\mathbb{M}(\hat{\mathcal{B}})$ the set of Markovian bandit problems $M'$ that satisfy Equation 5.

**Theorem 3.** *For any algorithm that computes indices locally, there exists a Markovian bandit problem $M$, an initial state $\boldsymbol{x}$ and estimates $\hat{\mathcal{B}}^a := (\hat{\boldsymbol{r}}^a, \hat{Q}^a, b_a^r, b_a^Q)$ such that $M \in \mathbb{M}(\hat{\mathcal{B}})$ and*

$$\sup_{M' \in \mathbb{M}(\hat{\mathcal{B}})} V_{M'}^{\pi^{I(\hat{\mathcal{B}})}}(\boldsymbol{x}) < \sup_{\pi} V_M^{\pi}(\boldsymbol{x}).$$

*Proof.* The proof presented in Appendix C is obtained by constructing a set $\mathbb{M}$ and two MDPs $M_1$ and $M_2$ in $\mathbb{M}$ such that Equation 11 cannot hold simultaneously for both $M_1$ and $M_2$. □

This theorem implies that one cannot define local indices such that Equation 11 holds for all bandit problems $M \in \mathbb{M}_k$. Yet, the use of this inequality is central in the regret analysis of UCRL2 (see the proof of UCRL2 (Jaksch et al., 2010)). This implies that the current methodology to obtain regret bounds for UCRL2 and its variants, *e.g.,* Bourel et al. (2020); Fruit et al. (2018); Talebi & Maillard (2018); Filippi et al. (2010), that use Extended Value Iteration is not applicable to bound the regret of their modified version that computes indices locally.

Note that for any set $\mathbb{M}$ such that $M \in \mathbb{M}$, there still exists an index policy $\pi^{\mathrm{ind}}$ that is optimistic because all MDPs in $\mathbb{M}$ are Markovian bandit problems. This optimistic index policy satisfies

$$\sup_{M' \in \mathbb{M}} V_{M'}^{\pi^{\mathrm{ind}}} \geq \sup_{\pi} V_M^{\pi}.$$

This means that restricting to index policies is not a restriction for optimism. What Theorem 3 shows is that an optimistic index policy can be defined only after the most optimistic MDP $M \in \mathbb{M}$ is computed and computing optimistic policy and $M$ simultaneously depends on the confidence sets of all arms.

Therefore, we believe that UCRL2 and its variants cannot compute optimistic policy locally: they should all require the joint knowledge of all $\{\hat{\mathcal{B}}^a\}_{a \in [n]}$.

# 7 Numerical experiments

In complement to our theoretical analysis, we report, in this section, the performance of our three algorithms in a model taken from the literature. The model is an environment with 3 arms, all following a Markov chain that is obtained by applying the optimal policy on the river swim MDP. A detailed description is given in Appendix D, along with all hyperparameters that we used. Our numerical experiments suggest that MB-PSRL outperforms other algorithms in term of average regret and is computationally less expensive than other algorithms. To ensure reproducibility, the code and data of our experiments are available at https://gitlab.inria.fr/kkhun/learning-in-rested-markovian-bandit.

**Performance result** We investigate the average regret and policy computation time of each algorithm. To do so, we run each algorithm for 80 simulations and for $K = 3000$ episodes per simulation. We arbitrarily choose the discount factor $\beta = 0.99$. In Figure 1(a), we show the average cumulative regret of the 3 algorithms. We observe that the average regret of MB-UCBVI is larger than those of MB-PSRL and MB-UCRL2. Moreover, we observe that MB-PSRL obtains the best performance and that its regret seems to grow slower than $O(\sqrt{K})$. This is in accordance to what was observed for PSRL (Osband et al., 2013). Note that the expected number of time steps after $K$ episodes is $K/(1-\beta)$ which means that in our setting with $K = 3000$ episodes there are $300\,000$ time steps in average. In Figure 1(b), we compare the computation time of the various algorithms. We observe that the computation time (the $y$-axis is in log-scale) of MB-PSRL

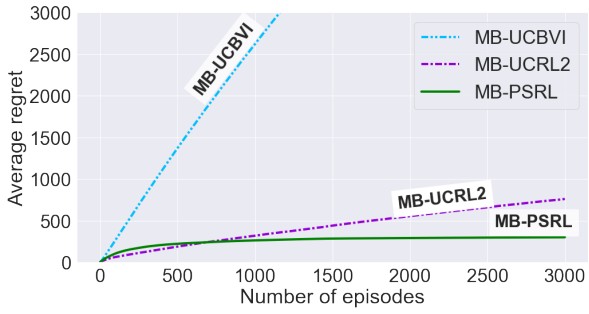
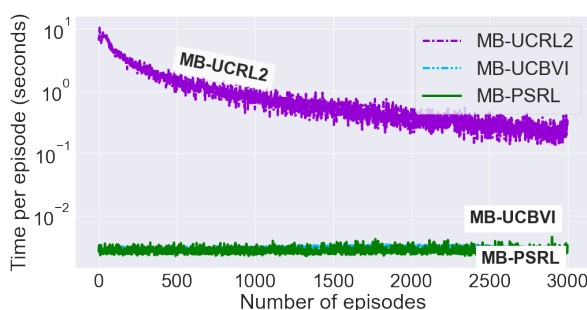

(a) Average cumulative regret in function of the number of episodes.

(b) Average runtime per episode. The vertical axis is in log-scale.

Figure 1: Experimental result for the three 4-state random walk arms given in Table 1. The $x$-axis is the number of episodes. Each algorithm is identified by a unique color for all figures.

and MB-UCBVI, the index-based algorithms, are the fastest by far. Moreover, the computation time of these algorithms seem to be independent of the number of episodes. These two figures show that MB-PSRL has the smallest regret and computation time among all compared algorithms.

**Robustness (larger models and different priors)**  To test the robustness of MB-PSRL, we conduct two more sets of experiments that are reported in Appendix E. They confirm the superiority of MB-PSRL. The first experiment is an example from Duff (1995) with 9 arms each having 11 states. This model illustrates the effect of the curse of dimensionality: the global MDP has $11^9$ states which implies that the runtime of MB-UCRL2 makes it impossible to use, while MB-PSRL and MB-UCBVI take a few minutes to complete 3000 episodes. Also in this example, MB-PSRL seems to converge faster to the optimal policy than MB-UCBVI. The second experiment tests the robustness of MB-PSRL to the choice of prior distribution. We provide numerical evidences that show that, even when MB-PSRL is run with a prior $\phi$ that is not the one from which $M$ is drawn, the regret of MB-PSRL remains acceptable (around twice the regret obtained with a correct prior).

## 8 Conclusion

In this paper, we present MB-PSRL, a modification of PSRL for Markovian bandit problems. We show that its regret is close to the lower bound that we derive for this problem while its runtime scales linearly with the number of arms. Furthermore, and unlike what is usually the case, MB-PSRL does not have an optimistic counterpart that scales well: we prove that MB-UCRL2 also has a sub-linear regret but has a computational complexity exponential in the number of arms. This result generalizes to all the variants of UCRL2 that rely on extended value iteration. We nevertheless show that OFU approach may still be pertinent for Markovian bandit problem: MB-UCBVI, a version of UCBVI can use Gittins indices and does not suffer from the dimensionality curse: it has a sub-linear regret in terms of the number of episodes and number of arms as well as a linear time complexity. However its regret remains larger than MB-PSRL.

The broad implication of this work is that, on the one hand, if a weakly coupled MDP or factored MDP can be solved efficiently when all the parameters are known, then PSRL can be adapted to have efficient regret and runtime. On the other hand, solving weakly coupled MDP or factored MDP efficiently when all the parameters are known does not imply that all optimistic algorithms are computationally efficient. This is a major difference between the Bayesian and the optimistic approach.

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
