# OpenReview forum: "Learning Algorithms for Markovian Bandits:\\Is Posterior Sampling more Scalable than Optimism?"
_TMLR — Accepted by TMLR_

### Review · Reviewer_pt7A · 2022-07-17

**Summary Of Contributions:**

This paper proposes a general framework to solve the Markovian bandit problem and three algorithms under this framework. The authors provided regret guarantees for the three algorithms. The regret is linear in the size of the state space associated with each arm. If we naively apply the existing regret guarantees for MDP problems, the regret will be exponential. The authors also provided a lower bound as well as empirical results.

**Broader Impact Concerns:**

I don't see any concerns on the ethical implications of this work. So I don't think a Broader Impact Statement is needed.

**Requested Changes:**

As mentioned above, I would like to see a better discussion on the motivation of studying this problem.

Relatively minor changes:

The authors use “markovian” and "bayesian" in many places. It should be “Markovian” and "Bayesian"

Last paragraph on page 3: what is the difference between the symbol \mathcal{X} and \mathcal{E}?

First paragraph of Section 4: “The structure of the three MB-* algorithm is similar” -> “The structures of the three MD-* algorithms are similar”

I think it would be helpful to discuss the motivation of choosing geometrically distributed episode length.

**Strengths And Weaknesses:**

Strengths:

The claims in the paper are supported by the content. The authors claim that the three algorithms that they propose have regret guarantees that are linear in S and sublinear in K. The authors provided proof to support the results. I did not check the details but the theory part seems to be sound. The authors also claim that two algorithms MB-PSRL and MB-UCBVI are computationally scalable, whereas MB-UCRL2 is not. This is supported by empirical evidence. Overall, I think this is a solid work, the paper is overall well-written and easy to follow.

Weaknesses:

After reading the paper, I did not get why Markovian bandit is an important problem. What is the practical motivation to study this problem? I think some discussions would be helpful.

---

> ### Author Response · Authors · 2022-07-22
> **Answer to Reviewer pt7A**
>
> We thank the reviewer for the positive and constructive comment. We answer the
> two questions below, and will incorporate this in the paper once all reviews
> are available.
>
> **Motivation**: Markovian bandit problem is a very natural problem and has many
> applications in scheduling (see for instance the book "Multi-armed bandit
> allocation indices » by Gittins et al. (2011) and the reference therein).  Learning
> algorithms for such problems have received quite some attention recently, most
> of them consider markovian restless bandit (Ortner et al., 2012; Jung & Tewari, 2019; Fu et al., 2019; Wang et al., 2020; Avrachenkov & Borkar, 2022) and some also
> consider the rested case (Duff, 1995) that we study in the paper. While the
> restless case is richer than the rested case, it is also harder and all of
> the cited papers only consider special classes of restless bandit problems.
> Moreover, they generally do not consider the computational complexity because
> restless bandit are in general intractable. Our study is the first to study
> learning algorithms for generic markovian rested bandit problems and we give a
> complete picture for both the performance and the computational complexity of a
> learning algorithms for markovian bandits.  To us, one of the most important
> contribution of the paper is the counter-intuitive fact that using « optimistic
> index » does not work, which leads MB-UCRL to be computationally expansive.
>
> **Episode length** The use of a Geometrically distributed episode length in this
> paper is because this allows us to compute an optimal policy by using Gittins
> index. Also, while our definition of regret is equivalent to other regret
> definition used in the literature (He et al., 2021), the randomness of episode
> length makes our setting more challenging then the fixed finite episode length
> setting. This demands new techniques to bound the regret (see Lemmas 1 and 4).

---

### Review · Reviewer_GzA5 · 2022-07-26

**Summary Of Contributions:**

This paper is concerned with Markov bandits. Three algorithms MB-PSRL, MB-UCRL2, and MB-UCBVI are proposed. Theoretical guarantees and empirical investigations are provided.

**Requested Changes:**

1. I'm not familiar with the specific Markovian bandit setting and probably I'm missing something, could you provide some motivating examples?

2. "its number of dimensions" is unclear in the introduction. I guess here you mean the number of state is exponential in some dimensions. And I think tabular algorithms just pays the complexity about the size of the state space. The proposed algorithm computationally also has a dependence in n, so I don't quite understand the argument here.

3. Some clarification is needed about Bayesian regret vs frequentist regret. Bayesian regret is a weaker guarantee. Recently there is a line of research that studies the frequentist regret for Beyesian/posterior sampling RL algorithms. The refs also seems to be outdated. See the below ref and the refs therein.

Refs:
Worst-Case Regret Bounds for Exploration via Randomized Value Functions
Improved worst-case regret bounds for randomized least-squares value iteration
Randomized Exploration for Reinforcement Learning with General Value Function Approximation

4. I think MB-UCBVI should have a better bound, i.e., a root S bound which matches the lower bound. Could you comment on why the S dependence is loose in MB-UCBVI? This is different from the optimal S dependence in UCBVI.

5. Could you comment on why there is no similar length of horizon factor (H) appearing in the bound? I think the problem setting is also sort of episodic, and in the standard MDP we would have H dependence.

6. The beginning of the related work is not clear. It has an informal and vague description about the setting and I feel it does not help. I still do not see the definition after reading that. Maybe just drop this part since you formally introduce it later.

7. In Alg1, I think the update of policy pi is missing. Probably add a line and pointers to the specific equations afterwards.

8. “The situation is radically different when considering the processing time: the runtime of MB-PSRL is linear in the number of arms, while the runtime of MB-UCRL2 is exponential in n”   It looks like n is not the number of arms, but it turns out yes. Probably be consistent.

9. "Hence, MB-PSRL and MB-UCBVI successfully escape from the curse of dimensionality." Can you comment on what they mean specifically?

10. I didn't check the proof, so no comment about its correctness.

**Strengths And Weaknesses:**

Strengths:
Markovian bandits problem is interesting. But I’m not familiar with the literature in this specific setting.


Weaknesses:
Presentation could be improved.
Literature/references seems to be outdated.

---

> ### Author Response · Authors · 2022-09-05
> **Our answer to reviewer's questions**
>
> 1. Markovian bandit have been applied to many problems such as single-machine scheduling, choosing a job, industrial research problem... More applications can be found in Section 3.6 of Puterman (2014) and Gittins et al. (2011).  Note that our paper is not motivated by a particular application and this is why we do not discuss such an application in detail in the paper.  Rather, our aim is to compare the Bayesian approach and the optimistic approach in the special case of weakly coupled MDPs for which there exists an efficient algorithm to compute the optimal policy.
>
> 2. As the reviewer says, we mean that the state space is exponential in the number of arms (which we wrongly phrase as the number of dimensions).  Hence, classical algorithms pay a complexity that is exponential in the number of arms whereas an algorithm like MB-PSRL or MB-UCBVI has a complexity linear in the number of arms.
>
> 3. It is true that, for PSRL, we use the notion of Bayesian regret. Obtaining a frequentist regret bound would be a stronger results but we believe that such a result is difficult to obtain.  There are two main reasons for that:
> i. Our algorithm is PSRL-like. We are aware of works that deal with  the worst-case regret for Thompson Sampling algorithms like the work of Agrawal and Goyal (2017) in stochastic bandit and Hamidi and Bayati (2020) in linear bandit. We are not aware of any work that considers the worst-case regret for PSRL (we do not believe that such a bound exists but if the reviewer has references to suggest, we would be happy to include them).
> ii. There exist PSRL-like algorithms that have frequentist guarantees but they are not purely Bayesian. They are a combination of the optimistic and Bayesian approaches. According to our impossibility results on optimistic approaches, we believe that existing optimistic posterior sampling algorithms are not scalable in Markovian bandit problems as their optimism is considered on the global value function (ie, the value function of the MDP whose state size is exponential in the number of arms). The line of research pointed by the reviewer is interesting and the references are about Randomized Least Squares Value Iteration (RLSVI). Basically, RLSVI functions by randomizing the empirical value function. Applying RLSVI to Markovian bandit problem means adding carefully chosen noise to the global empirical value function to create randomized state action value function whose size is $nS^n$ where $n$ is the number of arms and $S$ is the state size per arm. This means that RLSVI is not scalable (at least in terms of memory storage, let alone the operation cost to deduce a policy from the aforementioned state action value function). Note that the main avantage of MB-PSRL and MB-UCBVI algorithms over the generic ones is to avoid working on the global problem but to work on arms independently from one arm to another arm.
>
> Note also that our regret lower bound in Theorem 2 is a Bayesian regret lower bound. While the worst-case regret guarantee is stronger then Bayesian regret guarantee, the Bayesian minimax regret lower bound is stronger then the worst-case regret lower bound.
>
> 4. The factor $S$ (and not $\sqrt{S}$) is not a typo and we do not think that it can be easily improved.
>
> *More details:* To prove that UCBVI has a root S dependence (and not S), the authors of UCBVI define what they call a "correction" term that crucially relies on the fact that $\bar{V}$ is an upper bound of the real value function $V$.  In our setting, however, we cannot use the same trick because of the discounted nature of the problem and the use of the value functions $W$ that is dependent on a random episode length. In this setting, $\bar{W}$ is not necessarily an upper bound of $W$.  So, the approach to deal with the correction term as done in Step 1 page 6 of UCBVI cannot be adapted to MB-UCBVI. We are therefore left with a $S$ (and not $\sqrt{S}$) factor and we do not believe that this can be easily improved.  We will integrate this answer in the new version of our paper and add detail discussion about the dependence on $S$ in Appendix A.5.4.
>
> 5. In our discounted setting, the average length of an episodes is $1/(1-\beta)$, where $\beta$ is the discount factor. In the main body of the paper, our (upper and lower) bounds on the regret include the terms in $1/(1-\beta)$ and therefore include the dependence on the horizon length.
>
> 6,8. Thank you for pointing out these typos. We reformulated the corresponding sections.
>
> 9. We wanted to say "Hence, MB-PSRL and MB-UCBVI have a regret and a runtime both linear in the number of arms.". We reformulated these sentences in the new version.
>
> For 7. We are not to sure to understand what the reviewer means because Line 2 of the algorithm is "Compute a new policy".

---

### Review · Reviewer_UZTB · 2022-08-28

**Summary Of Contributions:**

This paper studies Markovian bandit problems where
 a) arm state transitions are restful;
 b) rewards come with an expoential discounting factor $\beta$;
 c) the time-horizen length in each episode is i.i.d. geometrically distributed with parameter $1-\beta$.

In the paper, three new algorithms MB-PSRL, MB-UCRL and MB-UCBVI are presented, all achieving $\tilde O(S\sqrt {nK})$ regret, which is nearly optimal up to an $\sqrt S$ factor compared to a Bayesian minimax regret lower-bound given in the paper. Among the three proposed algorithms, MB-PSRL and MB-UCBVI can be implemented efficiently.



**Requested Changes:**

The paper's approach heavily relies on exact planning via Gittins indices, which further heavily relies on the equivalence between discounted infinite time-horizons and finite geometric time horizons with a matching parameter. If the setting is slightly changed (e.g., all episodes have fixed uniform time-horizon lengths, or they are still geometrically i.i.d. but the parameter is no longer $1-\beta$), then Gittins indices policy is no longer optimal. I would like to see more discussion this.

Due to the above reason, I am afraid it is not very fair to claim advantages over prior works on finite-horizon settings when making comparisons in the literature review section, they are just different settings.

Finally, I think it is better to emphasize more on the regret guarantee of MB-PSRL in the abstract and introduction section. The presented regret upper-bound of MB-PSRL is for Bayasian regret, not for worse-case regret. It is an interesting open problem to find a prior for MB-PSRL so that the posterier can be efficiently maintained (e.g., having conjugate distributions) and upon which we can also develop worst-case style regret guarantees.

**Strengths And Weaknesses:**

Strength
========
The studied setting (restful bandit, reward discounting factor matching with the geometric time-horizon length) allows to efficiently compute the exact optimal policy when problem-instance parameters are completely known, by using the celebrated Gittins indices policy. Since we are able to efficiently solve the planning version problems, plugging the exact planning solutions to common UCB-optimistic frameworks and TS posterior-sampling frameworks leads to the proposed new algorithms. Hence the overall picture of this paper is clear and intuitive. The key proofs seem rigorous and convincing.

The impossibility result in Section 6.2 is interesting. It does rule out some candidates for optimistic MB algorithms (computing local indices before picking an optimistic problem parameter instance won't have desired optimistic properties).

Weekness and Concern
=====================
The paper's approach heavily relies on exact planning via Gittins indices, which further heavily relies on the equivalence between discounted infinite time-horizons and finite geometric time horizons with a matching parameter. If the setting is slightly changed (e.g., all episodes have fixed uniform time-horizon lengths, or they are still geometrically i.i.d. but the parameter is no longer $1-\beta$), then Gittins indices policy is no longer optimal. I would like to see more discussion this. Intuitively, as long as the rewards are discounted, when time-horizon lengths are sufficiently large, Gittins indices policy should be very close to the optimal policy, so maybe the regret guarantees of MB-PSRL and MB-UCBVI can still hold after carefully analyzing the error between Gittins indices policy and OPT.

Due to the above reason, I am afraid it is not very fair to claim advantages over prior works on finite-horizon settings when making comparisons in the literature review section, they are just different settings.

Finally, I think it is better to emphasize more on the regret guarantee of MB-PSRL in the abstract and introduction section. The presented regret upper-bound of MB-PSRL is for Bayasian regret, not for worse-case regret. It is an interesting open problem to find a prior for MB-PSRL so that the posterier can be efficiently maintained (e.g., having conjugate distributions) and upon which we can also develop worst-case style regret guarantees.

---

> ### Author Response · Authors · 2022-09-05
> **Our answer about the dependence on Gittins index policy, avantage over finite horizon, and worst-case regret**
>
> # About the special Gittins policy
> We use Gittins index as leverage on the fact that for someone who know all the parameters, there exists an efficient algorithm to compute the optimal policy. The regret bounds that we obtain do not depend on the use of such a policy (for any weakly coupled or factored MDP, one could do a similar proof).  What the existence of an easily computable policy does is that it makes MB-PSRL and MB-UCBVI computationally efficient.  This does not relies on the fact that this optimal policy is Gittins index policy but just that it is efficiently computable. This does not work for optimistic  algorithms like MB-UCRL2 that build confidence sets for transition probabilities and therefore need to rely on Extended Value Iteration (or its variant) to guarantee the optimism.
>
> Our conclusion: On the one hand, if the problem can be solved efficiently, then the Bayesian approach (we mean purely Bayesian and exclude optimistic posterior sampling) is efficient both in terms of learning and computation.  On the other hand, solving the known problem computationally efficient is not a sufficient condition for all optimistic algorithms to be computationally efficient (we will discuss that in the conclusion of the paper).
>
> # Possible advantage over finite horizon.
>
> We are not claiming that our method provides an advantage over prior work on finite-horizon: our technique would not bring anything to the classical finite horizon setting. Rather, what we want to say is that having geometrically distributed horizon requires a special treatment that is not needed for finite horizon setting.
>
> # Bayesian regret
>
> Indeed, our results for PSRL are only for the Bayesian regret. We emphasize this in the abstract/introduction (see also our answer to reviewer GzA5). We also think that finding a prior that would give a good worst-case regret bound is an interesting question. In fact, this would also be an interesting question for the classical PSRL.
>
> In addition to deriving algorithms that have guaranteed learning performance, it is fascinating for us to compare Bayesian approach with frequentist approach. This interest is also reflected in the title of our paper. In current literature, the metric for comparison is the regret or sample complexity (as in factored MDPs). Both quantities tell us how long the algorithms would take to interact with the environment in order to identify the optimal solution. We believe that computational complexity (the time that machine takes to compute a strategy or a policy) is an equally interesting metric for the comparison, especially for the implementation purpose. For this investigation, we think that the Markovian bandit problem is one of the best suited studying material because this problem is a large MDP whose optimal policy can be computed relatively fast when all the parameters are completely known (that is, Gittins index policy is an optimal policy that can be computed faster than using Value Iteration or Policy Iteration on the large MDP). In contrast, we are not aware of any efficient method to compute an optimal policy for general Restless Markovian bandit nor factored MDP. While Value and Policy iterations or their variants can compute an optimal policy for Restless bandit and factored MDP, they are intractable for large problems.  By studying Markovian bandit problems, we arrive to this insightful conclusion: On the one hand, if the problem can be solved computationally efficient, then Bayesian approach is efficient both in term of learning and computation. Also, existing work for computational purpose can be plugged into Bayesian approach when learning the unknown environment. On the other hand, we show that optimistic algorithms cannot always exploit the fact that the problem can be solved efficiently when all parameters are known.

---

### Decision · Action_Editors · 2022-10-18

**Recommendation:** Accept with minor revision

**Comment:**

The reviewers agree that the paper presents results that are supported by theoretical as well as empirical results. While two of the reviewers also consider the findings sufficiently interesting, one is a bit skeptical about the relevance of the setting. In that respect I agree with the authors in that the amount of literature on Markovian bandits shows that there is sufficient interest in research in that field and also provides several applications, that need not be further discussed in detail by the paper at hand.

However, the reviews also point out that the paper does not adequately discuss the limitations of the work and is rather sloppy when comparing different settings. Thus, while I reommend to accept the paper, in the final version the following minor revisions shall be made:
- Abstract as well as introduction shall point out the specific setting that is considered (i.e., rested bandits, episodic setting with a particular choice of the episodes).
- The term 'regret' (without any further specification) should not be used unless it is completely clear from context which precise notion of regret (e.g., Bayesian or expected) is considered. This holds in particular for the abstract where currently regret bounds with a different meaning are mentioned in the same sentence in quite a misleading manner.
- The limitations of the considered specific setting (as particularly pointed out by reviewer UZTB) should be adequately discussed.

Beside these important revisions I also noted that you cite the conference version (Auer et al 2008) instead of the extended journal version (Jaksch et al, JMLR 2010) of the UCRL2 paper.

**Audience:**

Although the paper considers a very specific problem setting, it contains results that will be interesting to researchers in reinforcement learning, in particular to those working on Markov bandits or are interested in the difference between optimistic vs posterior sampling methods.

**Claims And Evidence:**

The claims made in the paper are supported by theoretical results and complementary experiments.

---

> ### Author Response · Authors · 2022-11-10
> **Questions about camera-ready revision**
>
> Dear Action Editors,
>
> We have two questions related to Camera-ready revision:
> 1. What do we use between \usepackage[accepted]{tmlr} and \usepackage{tmlr}? Also, when we use \usepackage[accepted]{tmr}, some of us get a pdf with "Under review as submission to TMLR" as the header and some of us get "Published in Transactions on Machine Learning Research...".
> 2. What month and year shall we put for variable $\month$ and $\year$ in latex file?